# Effect of high and low risk susceptibles in the transmission dynamics of COVID-19 and control strategies

**Adnan Khan**[1], **Mohsin Ali**[1], **Wizda Iqbal**[2], **Mudassar Imran**[3]*

**1** Department of Mathematics, Lahore University of Management Sciences Opposite Sector 'U', DHA Lahore, Lahore, Pakistan, **2** National College of Business Administration & Economics, Lahore, Pakistan, **3** Department of Mathematics, Namal Institute Mianwali, Punjab, Pakistan

* mimran@asu.edu

**Data Availability Statement:** The data that support the findings of this study are openly available in at a. COVID-19 Data Repository by the Center for Systems Science and Engineering (CSSE) at Johns Hopkins University [https://github.com/

## Abstract

In this study, we formulate and analyze a deterministic model for the transmission of COVID-19 and evaluate control strategies for the epidemic. It has been well documented that the severity of the disease and disease related mortality is strongly correlated with age and the presence of co-morbidities. We incorporate this in our model by considering two susceptible classes, a high risk, and a low risk group. Disease transmission within each group is modelled by an extension of the SEIR model, considering additional compartments for quarantined and treated population groups first and vaccinated and treated population groups next. Cross Infection across the high and low risk groups is also incorporated in the model. We calculate the basic reproduction number $\mathcal{R}_0$ and show that for $\mathcal{R}_0 < 1$ the disease dies out, and for $\mathcal{R}_0 > 1$ the disease is endemic. We note that varying the relative proportion of high and low risk susceptibles has a strong effect on the disease burden and mortality. We devise optimal medication and vaccination strategies for effective control of the disease. Our analysis shows that vaccinating and medicating both groups is needed for effective disease control and the controls are not very sensitive to the proportion of the high and low risk populations.

## 1 Introduction

Coronavirus Disease (COVID-19) overshadowed all events in 2020 across the world and the pandemic is still ongoing in 2021. With the first case reported in Wuhan, China, in December 2019, the disease rapidly spread around the world, and was declared a pandemic by the WHO in March 2020 [1]. COVID-19 is caused by the SARS-CoV-2 virus which belongs to the family coronaviridae. Strains of this family were also responsible for the severe acute respiratory syndrome (SARS) and the Middle East respiratory syndrome (MERS) outbreaks in 2003 and 2012 [2].

COVID-19 is primarily spread by person to person contact through respiratory droplets. Symptoms appear 2-14 days after exposure and may include fever, dry cough, muscle pain, fatigue, and shortness of breath [3]. The symptoms are mild in 85% of the cases, and they vary

[CSSEGISandData/COVID-19] b. https://unstats.un.org/unsd/demographic-social/products/dyb/index.cshtml.

**Funding:** The author(s) received no specific funding for this work.

**Competing interests:** The authors have declared that no competing interests exist.

from severe in 10% to critical in 5% of those infected [2]. The severity and progression of COVID-19 are known to be exacerbated by the presence of co-morbidities such as diabetes, hypertension and cardio/cerebrovascular diseases [4]. It has also been observed that COVID-19 mortality risk is highly concentrated within the elderly population [5].

Mathematical models have found widespread use in the study of epidemics. The aim of such modelling is twofold, one to provide estimates of the severity of the outbreak by calculating quantities like the growth trends of the epidemic, estimates of the final outbreak size and duration of the outbreak and second to provide insights into efficacy of various control measures [6, 7]. Since the COVID-19 outbreak, several models have been proposed for the transmission dynamics and control of the disease. These include phenomenological models [8–10], which are useful at the beginning of an outbreak and mechanistic models which incorporate relevant and important transmission pathways [11–15]. For the first few months into the outbreak, the widely available control strategies were non-pharmaceutical, ranging from social distancing, usage of face masks, both of which reduce the effective contact rate to quarantine and isolation. Many studies have considered the effectiveness of these measures whereas some studies have also proposed optimal strategies using non-pharmaceutical measures [14, 16–19]. Since that time several treatments and a number of vaccines Pfizer-BioN-Tech, Moderna, AstraZeneca [20] have now either been approved or granted emergency approval.

The progression of COVID-19 has been markedly different in some countries. Starting in China, COVID-19 spread around the world rapidly, with Europe becoming the epicenter of the outbreak [2], followed by North and South America. With the first cases being reported in March 2020, Pakistan has had a very different epidemic curve as compared to China, Europe and the Americas, with a much lower disease burden and mortality. Many reasons have been suggested for this including, effective and early quarantine and isolation, a younger demographic and possibly difference in the prevalence of co-morbidities [21].

In this study, we propose an Ordinary Differential Equation (ODE) based compartmental model for the transmission dynamics of COVID-19. We have included compartments for high and low risk susceptible individuals to incorporate the role of demographics and co-morbidities in the progression of the disease and mortality. The disease transmission for both high and low risk populations is modelled by a variant of the SEIR model, with additional compartments representing quarantined, vaccinated and medicated population subgroups. Further, infection across the two groups is modelled by adding a cross-infection term to the force of infection. There are two main questions we investigate: first, does the proportion of high risk susceptibles explain the difference in the disease burden and/or mortality in different regions as described above, and second, if resources are limited, on which segment of the population, should the available control strategies be concentrated?

After describing the model, we derive some basic properties using standard dynamical systems theory. The system has two steady states, a disease free equilibrium (DFE), when the disease dies out in the long run and an endemic equilibrium (EE), where the disease is endemic in the population. We then determine a threshold quantity, the basic reproductive number $\mathcal{R}_0$ such that the DFE is stable whenever $\mathcal{R}_0 < 1$ and unstable otherwise, when $\mathcal{R}_0 > 1$ the EE is stable. Time series plots for different values of the high and low susceptible populations are plotted to explore how the disease burden and mortality varies with the varying proportion of these subgroups in the population. Next we explore different control measures that can be taken to reduce the disease burden. Using optimal control theory, efficient vaccine and medication strategies are devised, we also consider how the controls differ for the low and high risk groups. Finally, we summarize our findings in the conclusions section.

## 2 Effect of quarantine and medication

### 2.1 Model formulation

We propose a deterministic compartmental model for the transmission dynamics of COVID-19. The total population at any time instant, $N(t)$, is the sum of two sub-population groups, those at low risk for severe infection denoted by $N_L(t)$ and those at a higher risk denoted by $N_H(t)$. The transmission dynamics within each group are modelled by an extension of the SEIR model.

The susceptibles of Low-Risk $S_L(t)$ and High-Risk $S_H(t)$ groups are quarantined at rates $\theta_L(t)$ and $\theta_H(t)$ moving to the quarantine compartments $Q_L$ and $Q_H$. They can also move to the exposed class $E_L(t)$ and $E_H(t)$ after coming in contact with infected individuals, this occurs at rates $\beta_L, \beta_H$ for the low and high risk groups, respectively. In this model, we assume that the individuals in the low risk group have a higher contact rate with the infected population of that group as compared to the high risk group, mathematically, $\beta_L > \beta_H$. We also assume that exposed individuals are not infectious.

Exposed individuals move to the infected classes $I_L(t)$ and $I_H(t)$ at rates $\sigma_L$ and $\sigma_H$, it is assumed that the latency period is the same for both classes, $\frac{1}{\sigma_L} = \frac{1}{\sigma_H}$. Infected population(s) recover at rates $\gamma_L$ and $\gamma_H$, with $\gamma_L > \gamma_H$, this assumption follows from the fact that it takes longer to recover from a severe infection. A fraction of the infected individuals receive medication and move to the classes $M_L$ and $M_H(t)$ at rates $\tau_L$ and $\tau_H$. An Individual from $M_L$ and $M_H$ moves to the recovered classes at rates $\kappa_L$ and $\kappa_H$. The recovery rate for the low risk group with medication $\kappa_L$ is higher than that of the high risk medicated group $\kappa_H$.

An important feature of our model is the possibility of infection across the low and high risk groups. We assume that individuals from the low risk infected group can come into contact with the high risk susceptibles and vice versa, making cross infection possible. In fact, from very early on in the outbreak, there have been warnings about the low risk individuals not following social distancing protocols causing severe infection in the high risk population. We model this by assuming that the low risk infected population comes in contact with the high risk susceptibles at rate $\beta_{LH}$ and the high risk infected come in contact with the low risk susceptibles at a rate $\beta_{HL}$. It is also assumed that $\beta_L > \beta_{LH}$ and $\beta_{HL} > \beta_H$. This is based on the

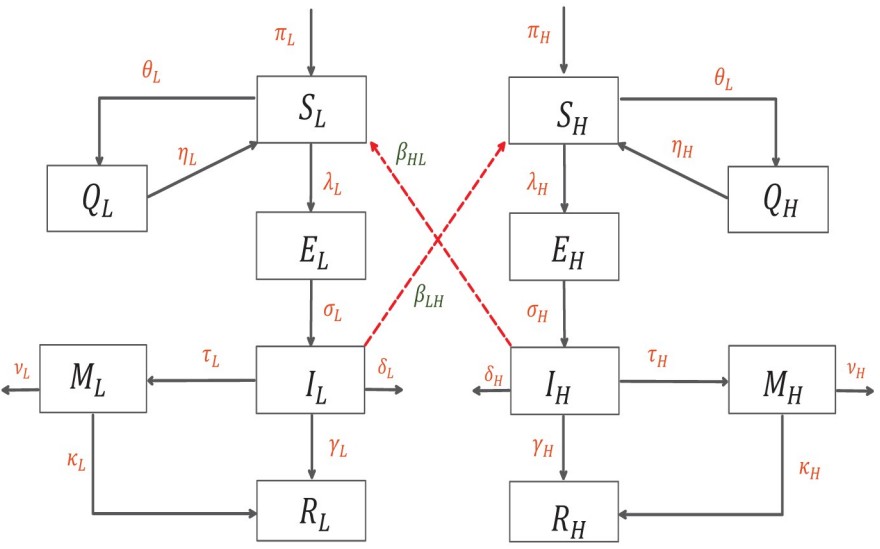

**Fig 1. Flow diagram of model (1).**

premise that high risk individuals are in general more cautious and observant of social distancing measures.

To summarize, the population is divided based on their risk for severe infection, transmission within each of these groups is then modelled by an extension of the SEIR model.

$$N(t) = N_L(t) + N_H(t)$$

where

$$N_L = S_L(t) + Q_L(t) + E_L(t) + I_L(t) + M_L(t) + R_L(t)$$
$$N_H = S_H(t) + Q_H(t) + E_H(t) + I_H(t) + M_H(t) + R_H(t)$$

The schematic of the transmission pathways is given in Fig 1 below.

## 2.2 Model equations

Mathematically, the model is described by the following system of Ordinary Differential Equations where the variables are described in Table 1.

$$
\begin{aligned}
\frac{dS_L}{dt} &= \pi_L - (\theta_L + \mu + \lambda_L)S_L + \eta_L Q_L \\
\frac{dQ_L}{dt} &= \theta_L S_L - (\mu + \eta_L)Q_L \\
\frac{dE_L}{dt} &= \lambda_L S_L - (\mu + \sigma_L)E_L \\
\frac{dI_L}{dt} &= \sigma_L E_L - (\mu + \tau_L + \delta_L + \gamma_L)I_L \\
\frac{dM_L}{dt} &= \tau_L I_L - (\mu + v_L + \kappa_L)M_L \\
\frac{dR_L}{dt} &= \gamma_L I_L + \kappa_L M_L - \mu R_L \\
\frac{dS_H}{dt} &= \pi_H - (\theta_H + \mu + \lambda_H)S_H + \eta_H Q_H \\
\frac{dQ_H}{dt} &= \theta_H S_H - (\mu + \eta_H)Q_H \\
\frac{dE_H}{dt} &= \lambda_H S_H - (\mu + \sigma_H)E_H \\
\frac{dI_H}{dt} &= \sigma_H E_H - (\mu + \tau_H + \delta_H + \gamma_H)I_H \\
\frac{dM_H}{dt} &= \tau_H I_H - (\mu + v_H + \kappa_H)M_H \\
\frac{dR_H}{dt} &= \gamma_H I_H + \kappa_H M_H - \mu R_H
\end{aligned}
\tag{1}
$$

where $\lambda_L$ and $\lambda_H$ respectively are force of infection for low and high risk groups

$$\lambda_L = \frac{\beta_L}{N_L}(I_L + \phi_L M_L) + \frac{\beta_{HL}I_H}{N_L} \tag{2}$$

$$\lambda_H = \frac{\beta_H}{N_H}(I_H + \phi_H M_H) + \frac{\beta_{LH}I_L}{N_H} \tag{3}$$

**Table 1. Description of the variables of the model (1) and (8).**

| Variable | Description |
|----------|-------------|
| $N_L$ | Total population of individuals at Low Risk |
| $N_H$ | Total population of individuals at High Risk |
| $S_L$ | Susceptible individuals for Low Risk |
| $S_H$ | Susceptible individuals for High Risk |
| $Q_L$ | Susceptible individuals Quarantined at Low Risk |
| $Q_H$ | Susceptible individuals Quarantined at High Risk |
| $E_L$ | Individuals Exposed to corona virus at Low Risk |
| $E_H$ | Individuals Exposed to corona virus at High Risk |
| $I_L$ | Individuals Infected with corona virus at Low Risk |
| $I_H$ | Individuals Infected with corona virus at High Risk |
| $M_L$ | Medication for infected/susceptible individuals at Low Risk |
| $M_H$ | Medication for infected/susceptible individuals at High Risk |
| $R_L$ | Susceptible individuals Recovered from virus at Low Risk |
| $R_H$ | Susceptible individuals Recovered from virus at High Risk |
| $V_L$ | Vaccinated of Low risk individuals |
| $V_H$ | Vaccinated of High risk individuals |

## 2.3 Basic properties

Model (1) has non-negative time series solutions for non-negative initial conditions. i.e. the differential system is well posed and bounded in positive orbit for all $t \geq 0$ with non-negative initial values.

**Lemma 2.1**. *For a given non-negative initial conditions of state variables, there exists a unique solution $S_L$, $Q_L$, $E_L$, $I_L$, $M_L$, $R_L$, $S_H$, $Q_H$, $E_H$, $I_H$, $M_H$, $R_H$ respectively, for all time $t \geq 0$. Moreover, The closed set*:

$$
\mathcal{D} = \left\{ (S_L, Q_L, E_L, I_L, M_L, R_L, S_H, Q_H, E_H, I_H, M_H, R_H) \in \mathbb{R}_+^{12} : \right.
$$

$$
\left. S_L + Q_L + E_L + I_L + M_L + R_L + S_H + Q_H + E_H + I_H + M_H + R_H \leq \frac{\pi_L + \pi_H}{\mu} \right\}
$$

*is positively invariant.*

Proof is attached in the Appendix A.

## 2.4 Steady state analysis

**2.4.1 Disease free equilibrium (DFE).** The model (1) attains the disease free equilibrium state when there is no force of infection i.e. $\lambda_L$ (2) and $\lambda_H$ (3) are zero. Let $\mathcal{E}_0$ denote the DFE of the model.

$$
\begin{aligned}
\mathcal{E}_0 &= (S_L^*, Q_L^*, E_L^*, I_L^*, M_L^*, R_L^*, S_H^*, Q_H^*, E_H^*, I_H^*, M_H^*, R_H^*) \\
&= \left( \frac{\pi_L(\mu + \eta_L)}{\mu(\mu + \eta_L + \theta_L)}, \frac{\pi_L \theta_L}{\mu(\mu + \eta_L + \theta_L)}, 0, 0, 0, 0, \frac{\pi_H(\mu + \eta_H)}{\mu(\mu + \eta_H + \theta_H)}, \right. \\
&\qquad \left. \frac{\pi_H \theta_H}{\mu(\mu + \eta_H + \theta_H)}, 0, 0, 0, 0 \right)
\end{aligned}
\tag{4}
$$

The stability of disease free equilibrium is determined by a threshold quantity, the basic reproduction number $\mathcal{R}_0$.

**The basic reproduction number $\mathcal{R}_0$.** The Basic reproduction number $\mathcal{R}_0$ determines the average secondary infections produced by the single infected in a completely susceptible population. This is a measure of propagation of the infection in the population and can be used for inference about the extinction or endemicity of the infection in the population. The next generation operator method described by [22] is used to calculate $\mathcal{R}_0$, which is determined by the spectral radius of $FV^{-1}$, where $F$ (The New infection Matrix) and $V$ (Transmission Matrix) and are given below.

$$F = \begin{pmatrix} 0 & \beta_L \Omega_L & \beta_L \Omega_L \phi_L & 0 & \beta_{HL} \Omega_L & 0 \\ 0 & 0 & 0 & 0 & 0 & 0 \\ 0 & 0 & 0 & 0 & 0 & 0 \\ 0 & \Omega_H \beta_{LH} & 0 & 0 & \beta_H \Omega_H & \beta_H \Omega_H \\ 0 & 0 & 0 & 0 & 0 & 0 \\ 0 & 0 & 0 & 0 & 0 & 0 \end{pmatrix},$$

$$V = \begin{pmatrix} k_1 & 0 & 0 & 0 & 0 & 0 \\ -\sigma_L & k_2 & 0 & 0 & 0 & 0 \\ 0 & -\tau_L & k_3 & 0 & 0 & 0 \\ 0 & 0 & 0 & k_4 & 0 & 0 \\ 0 & 0 & 0 & -\sigma_H & k_5 & 0 \\ 0 & 0 & 0 & 0 & -\tau_H & k_6 \end{pmatrix}$$

where $\Omega_L = \dfrac{S_L^*}{N_L^*}$ and $\Omega_H = \dfrac{S_H^*}{N_H^*}$, $k_1 = \sigma_L + \mu$, $k_2 = \delta_L + \tau_L + \gamma_L + \mu$, $k_3 = \mu + v_L + \kappa_L$, $k_4 = \sigma_H + \mu$, $k_5 = \delta_H + \tau_H + \gamma_H + \mu$, $k_6 = \mu + v_H + \kappa_H$,

The basic reproductive number $\mathcal{R}_0 = \rho(FV^{-1})$ can be written as

$$\mathcal{R}_0 = \max\left( \frac{(A+B) - \sqrt{(A-B)^2 + 4C}}{2}, \frac{(A+B) + \sqrt{(A-B)^2 + 4C}}{2} \right) \tag{5}$$

$$= \frac{(A+B) + \sqrt{(A-B)^2 + 4C}}{2} \tag{6}$$

Where $A = \dfrac{\beta_H \sigma_H \Omega_H (\tau_H \phi_H + k_6)}{k_4 k_5 k_6}$, $B = \dfrac{\beta_L \sigma_L \Omega_L (k_3 + \tau_L \phi_L)}{k_1 k_2 k_3}$, $C = \dfrac{\sigma_H \Omega_H \beta_{HL} \sigma_L \Omega_L \beta_{LH}}{k_1 k_2 k_4 k_5}$

**Lemma 2.2.** [22] *The steady state (DFE) $\mathcal{E}_0$ of the model* (1) *is locally-asymptotically stable if $\mathcal{R}_0 < 1$, and unstable if $\mathcal{R}_0 > 1$.*

**2.4.2 Endemic equilibrium.** The model (1) attains the endemic equilibrium when $\lambda_L$ (2) and $\lambda_H$ (3) are non zero. Let $\mathcal{E}_1$ represent the endemic equilibrium of the model (1).

$$\mathcal{E}_1 = (S_L^{**}, Q_L^{**}, E_L^{**}, I_L^{**}, M_L^{**}, R_L^{**}, S_H^{**}, Q_H^{**}, E_H^{**}, I_H^{**}, M_H^{**}, R_H^{**}) \tag{7}$$

Moreover, the force of infection $\lambda_L$ and $\lambda_H$ can be written in terms of the endemic equilibrium

as

$$\lambda_L^{**} = \frac{\beta_L}{N_L^{**}}(I_L^{**} + \phi_L M_L^{**}) + \frac{\beta_{HL} I_H^{**}}{N_L^{**}}$$

$$\lambda_H^{**} = \frac{\beta_H}{N_H^{**}}(I_H^{**} + \phi_H M_H^{**}) + \frac{\beta_{LH} I_L^{**}}{N_H^{**}}$$

with $N_L^{**} = S_L^{**} + Q_L^{**} + E_L^{**} + I_L^{**} + M_L^{**} + R_L^{**}$ and $N_H^{**} = S_H^{**} + Q_H^{**} + E_H^{**} + I_H^{**} + M_H^{**} + R_H^{**}$

Solving for the transmission (1) at this specific fixed point, the endemic equilibrium becomes

$$S_L^{**} = \frac{\pi_L(\eta_L + \mu)}{\eta_L(\lambda_L^{**} + \mu) + \mu(\theta_L + \lambda_L^{**} + \mu)}, \quad Q_L^{**} = \frac{\pi_L \theta_L}{\eta_L(\lambda_L^{**} + \mu) + \mu(\theta_L + \lambda_L^{**} + \mu)},$$

$$E_L^{**} = \frac{\pi_L \lambda_L^{**}(\eta_L + \mu)}{k_1(\eta_L(\lambda_L^{**} + \mu) + \mu(\theta_L + \lambda_L^{**} + \mu))}, \quad I_L^{**} = \frac{\pi_L \lambda_L^{**}(\eta_L + \mu)\sigma_L}{k_1 k_2(\eta_L(\lambda_L^{**} + \mu) + \mu(\theta_L + \lambda_L^{**} + \mu))},$$

$$M_L^{**} = \frac{\pi_L \lambda_L^{**}(\eta_L + \mu)\sigma_L \tau_L}{k_1 k_2 k_3(\eta_L(\lambda_L^{**} + \mu) + \mu(\theta_L + \lambda_L^{**} + \mu))},$$

$$R_L^{**} = \frac{\pi_L \lambda_L^{**}(\eta_L + \mu)\sigma_L(k_3 \gamma_L + \kappa_L \tau_L)}{k_1 k_2 k_3(\eta_L(\lambda_L^{**} + \mu) + \mu(\theta_L + \lambda_L^{**} + \mu))},$$

$$S_H^{**} = \frac{\pi_H(\eta_H + \mu)}{\eta_H(\lambda_H^{**} + \mu) + \mu(\theta_H + \lambda_H^{**} + \mu)}, \quad Q_H^{**} = \frac{\pi_H \theta_H}{\eta_H(\lambda_H^{**} + \mu) + \mu(\theta_H + \lambda_H^{**} + \mu)},$$

$$E_H^{**} = \frac{\pi_H \lambda_H^{**}(\eta_H + \mu)}{k_4(\eta_H(\lambda_H^{**} + \mu) + \mu(\theta_H + \lambda_H^{**} + \mu))}, \quad I_H^{**} = \frac{\pi_H \lambda_H^{**}(\eta_H + \mu)\sigma_H}{k_4 k_5(\eta_H(\lambda_H^{**} + \mu) + \mu(\theta_H + \lambda_H^{**} + \mu))},$$

$$M_H^{**} = \frac{\pi_H \lambda_H^{**}(\eta_H + \mu)\sigma_H \tau_H}{k_4 k_5 k_6(\eta_H(\lambda_H^{**} + \mu) + \mu(\theta_H + \lambda_H^{**} + \mu))},$$

$$R_H^{**} = \frac{\pi_H \lambda_H^{**}(\eta_H + \mu)\sigma_H(k_6 \gamma_H + \kappa_H \tau_H)}{k_4 k_5 k_6(\eta_H(\lambda_H^{**} + \mu) + \mu(\theta_H + \lambda_H^{**} + \mu))}$$

**2.4.3 Numerical simulations.** Numerical Simulations are performed with the help of Matlab(ODE 45) using the parameter values given in the Table 2. Fig 2 shows the time series solutions of model (1). Solutions achieve the DFE and Endemic Equilibrium whenever the threshold quantity $\mathcal{R}_0$ is less than one and more than one, respectively. These results are in line with the qualitative results found above.

One of the issues we investigate is the dependence of disease burden on the proportion of high and low risk susceptibles in the population. As noted in the introduction, the epidemic curve has been very different in many South Asian countries as compared to Europe and America. One plausible explanation could be the difference in the numbers of high and low risk individuals based on demographics and perhaps co-morbidities in the populations. It is relatively easy to obtain the demographic data for different countries, data on the co-morbidities with COVID-19 is harder to unfold. Italy, which has a severe outbreak and very high mortality, has a high proportion of aging individuals, with around 23% of the population above the age of 65 years, whereas Pakistan has less than 5% of the population above 65. We plot in Fig 3, the time series for different proportion *f* of high risk individuals in the susceptible population, we look at the epidemic curve for *f* = 0.05, 0.1, 0.25 and 0.5.

It is clear from the graphs that the epidemic curve varies with the proportion of the high risk individuals *f*, not only is the maximum daily number of infected higher for a higher *f*, but the curve peaks later as well, both these factors contribute to a higher total infected as the

**Table 2. Description of the parameters of the model.**

| Parameters | Description | Values | |
|---|---|---|---|
| $\Pi_L$ | Recruitment rate for Humans at Low Risk | 10 | Assumed |
| $\Pi_H$ | Recruitment rate for Humans at High Risk | 10 | Assumed |
| $\mu$ | Natural death rate of humans at High/Low Risk | 60 years | Assumed |
| $\theta_L$ | Susceptible Quarantine rate of Susceptible individuals at Low Risk | 0.12 | Assumed |
| $\theta_H$ | Susceptible Quarantine rate of Susceptible individuals at High Risk | 0.15 | Assumed |
| $\eta_L$ | Waning rate of susceptible quarantined individuals at Low Risk | 1/28 | Assumed |
| $\eta_H$ | Waning rate of susceptible quarantined individuals at High Risk | 1/28 | Assumed |
| $\frac{1}{\sigma_L}$ | Incubation rate of susceptible individuals at Low Risk | 3−5 days | [13, 23] |
| $\frac{1}{\sigma_H}$ | Incubation rate of susceptible individuals at High Risk | 3−5 days | [13, 23] |
| $\tau_L$ | Medication rate of infected individuals at Low Risk | 0.1 | [13] |
| $\tau_L$ | Medication rate of infected individuals at High Risk | 0.1 | [13] |
| $\delta_L$ | Disease-induced death rate of individuals at Low Risk | 0.065 day$^{-1}$ | Estimated |
| $\delta_H$ | Disease-induced death rate of individuals at High Risk | 0.10 day$^{-1}$ | Estimated |
| $\beta_L$ | Effective contact rate | 0.8−1.5 | [13] |
| $\beta_H$ | Effective contact rate | 0.8−1.5 | [13] |
| $\frac{1}{\gamma_L}$ | Recovery rate of infected individuals at Low Risk | 10 days | [13, 23] |
| $\frac{1}{\gamma_H}$ | Recovery rate of infected individuals at High Risk | 14 days | [13, 23] |
| $\kappa_L$ | Recovery rate of quarantined individuals at Low Risk | 0.14 | [13, 23] |
| $\kappa_H$ | Recovery rate of quarantined individuals at High Risk | 0.14 | [13, 23] |
| $\beta_L H$ | Effective contact rate | 0.8−1.5 | [13] |
| $\beta_H L$ | Effective contact rate | 0.8−1.5 | [13] |

proportion of high risk individuals is increased. In our simulations we observe that over 120 days for $f = 0.05$, the total number of infected is 364,000, for $f = 0.1$, total infected are 428,000, $f = 0.25$ the total infected are around 566,000, for $f = 0.5$ the total infected are around 726,000.

Another major difference that has been observed in the COVID-19 outbreak is the low disease related morbidity in these countries as compared to Europe and the Americas. We explore whether this can be explained, at least to some degree, by the number of high risk individuals in a population. We plot in Fig 4, the cumulative deaths due to disease for different values of $f$ below.

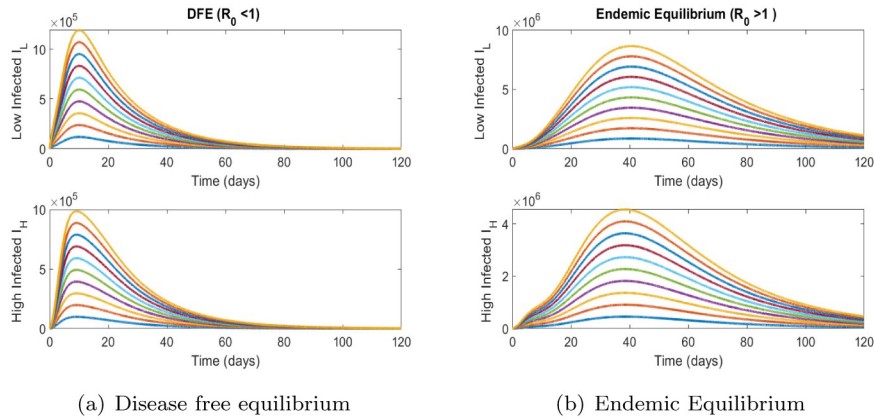

(a) Disease free equilibrium    (b) Endemic Equilibrium

**Fig 2. Time series simulations.** (a) Disease free equilibrium, (b) Endemic Equilibrium.

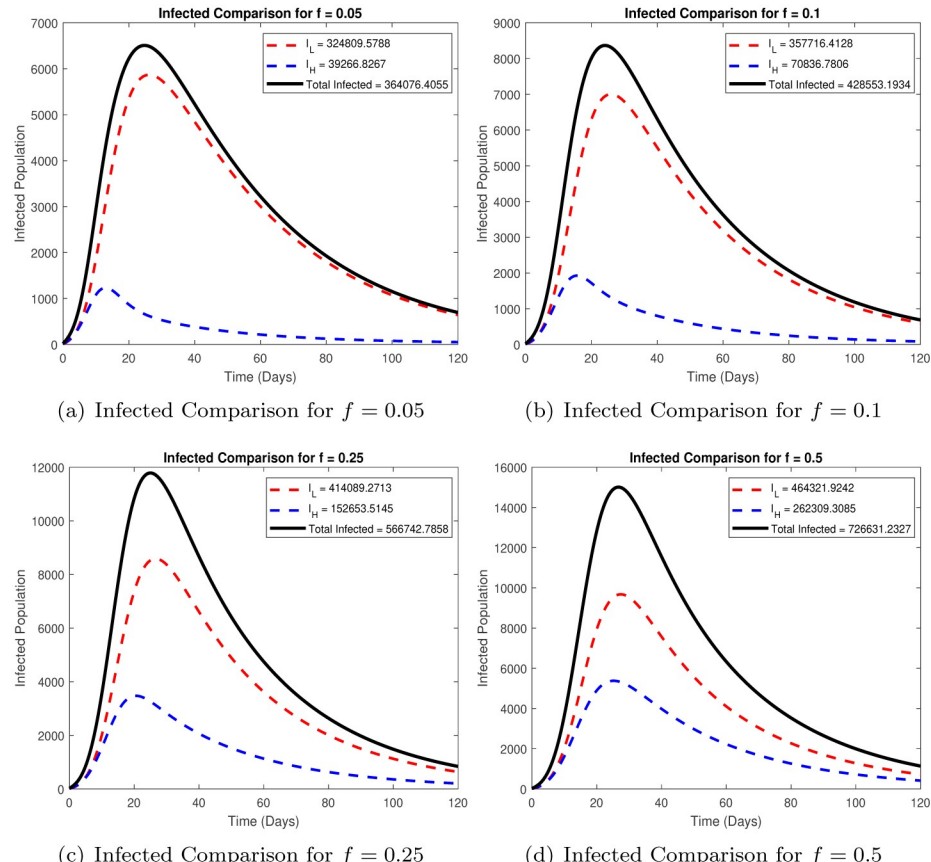

**Fig 3. Comparison of total infected for different values of $f$.** (a) Infected Comparison for $f$ = 0.05, (b) Infected Comparison for $f$ = 0.1, (c) Infected Comparison for $f$ = 0.25, (d) Infected Comparison for $f$ = 0.5.

We note that the disease mortality is significantly higher for a population with a greater proportion of high risk individuals. Over 120 days, for $f$ = 0.05 the total disease related deaths are around 25,000, for $f$ = 0.1 this number is around 21,000, for $f$ = 0.25 the total deaths due to disease are around 42,000 and for $f$ = 0.5 the total deaths due to disease are around 58,000. Our study establishes that both the disease burden and mortality is higher with a greater proportion of high risk individuals in the population.

We now look at the variation of $\mathcal{R}_0$ with different parameters of the model. To this end we plot in Fig 5, the contours of $\mathcal{R}_0$ varying two of the model parameters.

We note that contact rates for both risk classes need to be low in order to bring $\mathcal{R}_0$ less than one, to achieve this strict social distancing and masking protocols would need to be in place for both low and high risk individuals. Further, for lower quarantine rates we would need a high rate of medication in order to control the outbreak and vice versa, this translates into recommendation that both medication (which reduces the duration of the disease) and quarantine should be used together to control the epidemic.

As mentioned, at the beginning of the epidemic, non pharmaceutical interventions were the only control measures available, we now have several vaccines that have been approved for use against COVID-19. In the next section, we look at a variant of our model that incorporates the effects of vaccination.

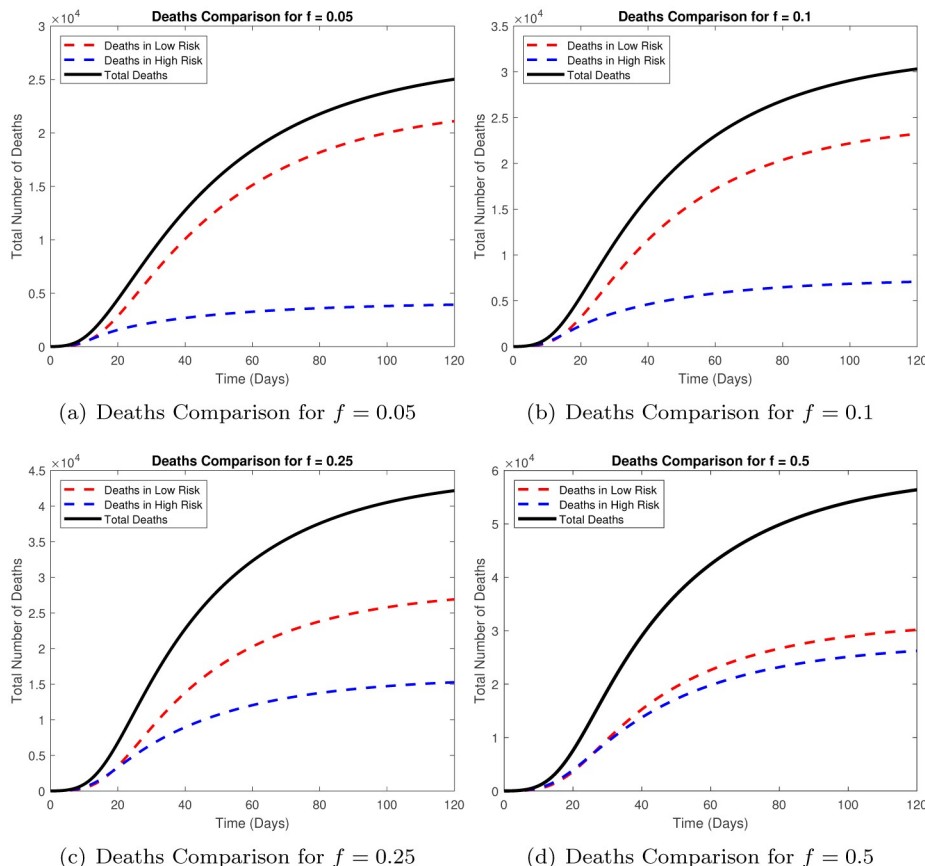

**Fig 4. Comparison of total deaths for different values of *f*.** (a) Deaths Comparison for *f* = 0.05, (b) Deaths Comparison for *f* = 0.1, (c) Deaths Comparison for *f* = 0.25, (d) Deaths Comparison for *f* = 0.5.

## 3 Effect of imperfect vaccine

In this section, we are interested in studying the effects of an imperfect vaccine on the transmission of the COVID-19. We consider that individuals are being vaccinated at rates $\xi_L$ and $\xi_H$ for the low risk and high risk classes respectively. There are several vaccines that are available at present, with vaccine effectiveness varying from 70% for AstraZeneca-University of Oxford

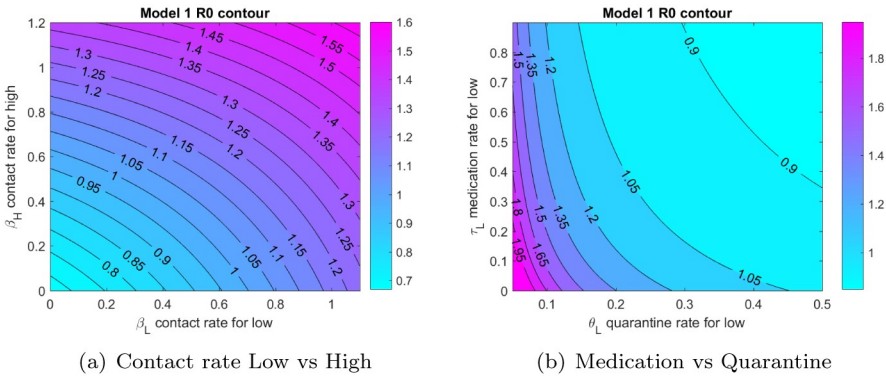

**Fig 5. Contours of $\mathcal{R}_0$.** (a) Contact rate Low vs High, (b) Medication vs Quarantine.

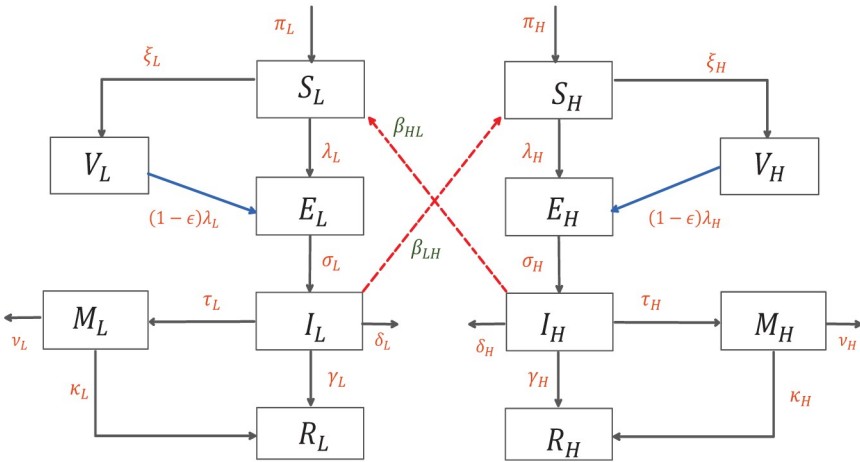

**Fig 6. Schematic diagram of imperfect vaccine transmission.**

to 95% for Pfizer pharma [20]. As a result, a small fraction of vaccinated individuals who are exposed to the COVID-19 virus eventually develop symptoms and become infected. Fig 6 describes the flow of transmission of the COVID-19 when an imperfect vaccine is available.

$$
\begin{aligned}
\frac{dS_L}{dt} &= \pi_L - (\xi_L + \mu + \lambda_L)S_L \\
\frac{dV_L}{dt} &= \xi_L S_L - \mu V_L - (1-\epsilon)\lambda_L V_L \\
\frac{dE_L}{dt} &= \lambda_L S_L + (1-\epsilon)\lambda_L V_L - (\mu + \sigma_L)E_L \\
\frac{dI_L}{dt} &= \sigma_L E_L - (\mu + \tau_L + \delta_L + \gamma_L)I_L \\
\frac{dM_L}{dt} &= \tau_L I_L - (\mu + v_L + \kappa_L)M_L \\
\frac{dR_L}{dt} &= \gamma_L I_L + \kappa_L M_L - \mu R_L \\
\frac{dS_H}{dt} &= \pi_H - (\xi_H + \mu + \lambda_H)S_H \\
\frac{dV_H}{dt} &= \xi_H S_H - \mu V_H - (1-\epsilon)\lambda_H V_H \\
\frac{dE_H}{dt} &= \lambda_H S_H + (1-\epsilon)\lambda_H V_H - (\mu + \sigma_H)E_H \\
\frac{dI_H}{dt} &= \sigma_H E_H - (\mu + \tau_H + \delta_H + \gamma_H)I_H \\
\frac{dM_H}{dt} &= \tau_H I_H - (\mu + v_H + \kappa_H)M_H \\
\frac{dR_H}{dt} &= \gamma_H I_H + \kappa_H M_H - \mu R_H
\end{aligned}
\tag{8}
$$

Where $\lambda_L$, $\lambda_H$ are defined as in Eqs (2) and (3).

## 3.1 Positivity and invariance

The vaccine model (8) has non-negative time series solutions for non-negative initial conditions which implies that the system is well posed and bounded in the positive orbit starting with non negative initial data.

**Lemma 3.1**. *For any given, non-negative initial conditions of state variables of the model* (8), *there exists a unique solution* $S_L, V_L, E_L, I_L, M_L, R_L, S_H, V_H, E_H, I_H, M_H, R_H$ *respectively, for all time* $t \geq 0$. *Moreover, The closed set:*

$$\mathcal{D} = \left\{ (S_L, V_L, E_L, I_L, M_L, R_L, S_H, V_H, E_H, I_H, M_H, R_H) \in \mathbb{R}_+^{12} : \right.$$

$$\left. S_L + V_L + E_L + I_L + M_L + R_L + S_H + V_H + E_H + I_H + M_H + R_H \leq \frac{\pi_L + \pi_H}{\mu} \right\}$$

*is positively invariant.*

Proof is presented in appendix.

## 3.2 Steady states: Disease free equilibrium

The vaccine transmission model (8) achieves the disease free equilibrium state when the force of infection $\lambda_L$ (2) and $\lambda_H$ (3) are both zero. Let $\mathcal{E}_{vac}^0$ denote the DFE of the model (8).

## 3.3 Disease free equilibrium

$$\mathcal{E}_{vac}^0 = (S_L^\star, V_L^\star, E_L^\star, I_L^\star, M_L^\star, R_L^\star, S_H^\star, V_H^\star, E_H^\star, I_H^\star, M_H^\star, R_H^\star)$$

$$\mathcal{E}_{vac}^0 = \left( \frac{\pi_L}{(\mu + \xi_L)}, \frac{\xi_L \pi_L}{\mu(\mu + \xi_L)}, 0, 0, 0, 0, \frac{\pi_H}{(\mu + \xi_H)}, \frac{\xi_H \pi_H}{\mu(\mu + \xi_H)}, 0, 0, 0, 0 \right) \tag{9}$$

The threshold quantity (basic reproduction number $\mathcal{R}_0^{vac}$) for disease free equilibrium is determined by finding the $F$ (The New infection Matrix) and $V$ (The Transmission Matrix) as

$$F = \begin{pmatrix} 0 & \beta_L \Lambda_L & \beta_L \Lambda_L \phi_L & 0 & \beta_{HL} \Lambda_L & 0 \\ 0 & 0 & 0 & 0 & 0 & 0 \\ 0 & 0 & 0 & 0 & 0 & 0 \\ 0 & \Lambda_H \beta_{LH} & 0 & 0 & \beta_H \Lambda_H & \beta_H \Lambda_H \phi_H \\ 0 & 0 & 0 & 0 & 0 & 0 \\ 0 & 0 & 0 & 0 & 0 & 0 \end{pmatrix}$$

$$V = \begin{pmatrix} k_1 & 0 & 0 & 0 & 0 & 0 \\ -\sigma_L & k_2 & 0 & 0 & 0 & 0 \\ 0 & -\tau_L & k_3 & 0 & 0 & 0 \\ 0 & 0 & 0 & k_4 & 0 & 0 \\ 0 & 0 & 0 & -\sigma_H & k_5 & 0 \\ 0 & 0 & 0 & 0 & -\tau_H & k_6 \end{pmatrix}$$

where $\Lambda_L = 1 - \epsilon \frac{V_L^\star}{N_L^\star}$ and $\Lambda_H = 1 - \epsilon \frac{V_H^\star}{N_H^\star}$, $k_1 = \sigma_L + \mu$, $k_2 = \delta_L + \tau_L + \gamma_L + \mu$, $k_3 = \mu + \nu_L + \kappa_L$, $k_4 = \sigma_H + \mu$, $k_5 = \delta_H + \tau_H + \gamma_H + \mu$, $k_6 = \mu + \nu_H + \kappa_H$,

The stability of the $\mathcal{E}^0_{vac}$ is determined by the value of the $\mathcal{R}^{vac}_0 = \rho(FV^{-1})$.

$$\mathcal{R}^{vac}_0 = \max\left(\frac{(A_0 + B_0) - \sqrt{(A_0 - B_0)^2 + 4C_0}}{2}, \frac{(A_0 + B_0) + \sqrt{(A_0 - B_0)^2 + 4C_0}}{2}\right)$$

$$\mathcal{R}^{vac}_0 = \frac{(A_0 + B_0) + \sqrt{(A_0 - B_0)^2 + 4C_0}}{2}$$

(10)

Where $A_0 = \frac{\beta_H \sigma_H \Lambda_H (\tau_H \phi_H + k_6)}{k_4 k_5 k_6}$, $B_0 = \frac{\beta_L \sigma_L \Lambda_L (k_3 + \tau_L \phi_L)}{k_1 k_2 k_3}$, $C_0 = \frac{\sigma_H \Lambda_H \beta_{HL} \sigma_L \Lambda_L \beta_{LH}}{k_1 k_2 k_4 k_5}$

$\mathcal{R}^{vac}_0$ is the expected number of secondary infections by single infected in the completely susceptible population. If $\mathcal{R}^{vac}_0 < 1$, on average the new infections decrease with time and the number of infections will approach the disease free equilibrium. In this case, $\mathcal{E}^0_{vac}$ will be a stable equilibrium state. On the contrary, if $\mathcal{R}^{vac}_0 > 1$, on average new infections increase with time and the disease will tend towards the endemic equilibrium state.

$$\mathcal{E}^1_{vac} = (S^{\star\star}_L, V^{\star\star}_L, E^{\star\star}_L, I^{\star\star}_L, M^{\star\star}_L, R^{\star\star}_L, S^{\star\star}_H, V^{\star\star}_H, E^{\star\star}_H, I^{\star\star}_H, M^{\star\star}_H, R^{\star\star}_H)$$

(11)

**Lemma 3.2**. [22] *The steady state (DFE) $\mathcal{E}^{vac}_0$ of the model* (8) *is locally-asymptotically stable if $\mathcal{R}^{vac}_0 < 1$, and unstable if $\mathcal{R}^{vac}_0 > 1$.*

### 3.4 Steady states: Endemic equilibrium

The endemic equilibrium is attained when the force of infection is not zero. i.e. $\lambda_i \neq 0$. $\mathcal{E}^{vac}_1$ represents the endemic equilibrium of the model (8)

$$\mathcal{E}^1_{vac} = (S^{\star\star}_L, V^{\star\star}_L, E^{\star\star}_L, I^{\star\star}_L, M^{\star\star}_L, R^{\star\star}_L, S^{\star\star}_H, V^{\star\star}_H, E^{\star\star}_H, I^{\star\star}_H, M^{\star\star}_H, R^{\star\star}_H)$$

(12)

where

$$S^{\star\star}_L = \frac{\pi_L}{\lambda_L + \xi_L + \mu}, \qquad V^{\star\star}_L = \frac{\xi_L \pi_L}{(\lambda_L + \xi_L + \mu)((1-\epsilon)\lambda_L + \mu)}$$

$$E^{\star\star}_L = \frac{\pi_L \lambda_L ((1-\epsilon)\lambda_L + (1-\epsilon)\xi_L + \mu)}{(\lambda_L + \xi_L + \mu)((1-\epsilon)\lambda_L + \mu)k_1},$$

$$I^{\star\star}_L = \frac{\pi_L \lambda_L \sigma_L ((1-\epsilon)\lambda_L + (1-\epsilon)\xi_L + \mu)}{(\lambda_L + \xi_L + \mu)k_2((1-\epsilon)\lambda_L + \mu)k_1}$$

$$M^{\star\star}_L = \frac{\lambda_L \tau_L \pi_L ((1-\epsilon)\lambda_L + (1-\epsilon)\xi_L + \mu)\sigma_L}{(\lambda_L + \xi_L + \mu)k_2((1-\epsilon)\lambda_L + \mu)k_3 k_1}$$

$$R^{\star\star}_L = \frac{\sigma_L ((1-\epsilon)\lambda_L + (1-\epsilon)\xi_L + \mu)(\gamma_L k_3 + \tau_L \kappa_L)\pi_L \lambda_L}{(\lambda_L + \xi_L + \mu)k_2((1-\epsilon)\lambda_L + \mu)k_3 k_1 \mu}$$

$$S^{\star\star}_H = \frac{\pi_H}{\lambda_H + \xi_H + \mu}, \qquad V^{\star\star}_H = \frac{\xi_H \pi_H}{(\lambda_H + \xi_H + \mu)((1-\epsilon)\lambda_H + \mu)}$$

$$E^{\star\star}_H = \frac{\lambda_H \pi_H ((1-\epsilon)\lambda_H + (1-\epsilon)\xi_H + \mu)}{(\lambda_H + \xi_H + \mu)((1-\epsilon)\lambda_H + \mu)k_4},$$

$$I^{\star\star}_H = \frac{\lambda_H \pi_H \sigma_H ((1-\epsilon)\lambda_H + (1-\epsilon)\xi_H + \mu)}{(\lambda_H + \xi_H + \mu)k_5((1-\epsilon)\lambda_H + \mu)k_4}$$

$$M^{\star\star}_H = \frac{\lambda_H \tau_H \pi_H ((1-\epsilon)\lambda_H + (1-\epsilon)\xi_H + \mu)\sigma_H}{(\lambda_H + \xi_H + \mu)k_5((1-\epsilon)\lambda_H + \mu)k_6 k_4}$$

$$R^{\star\star}_H = \frac{\sigma_H ((1-\epsilon)\lambda_H + (1-\epsilon)\xi_H + \mu)(\gamma_H k_6 + \tau_H \kappa_H)\pi_H \lambda_H}{(\lambda_H + \xi_H + \mu)k_5((1-\epsilon)\lambda_H + \mu)k_6 k_4 \mu}$$

Here, the force of the infection can be written with endemic equilibrium values as

$$\lambda_L^{\star\star} = \frac{\beta_L}{N_L^{\star\star}}(I_L^{\star\star} + \phi_L M_L^{\star\star}) + \frac{\beta_{HL}I_H^{\star\star}}{N_L^{\star\star}}$$

$$\lambda_H^{\star\star} = \frac{\beta_H}{N_H^{\star\star}}(I_H^{\star\star} + \phi_H M_H^{\star\star}) + \frac{\beta_{LH}I_L^{\star\star}}{N_H^{\star\star}}$$

We now plot, Fig 7, the epidemic curve for different values of the model parameters.

We note that the results follow the qualitative analysis presented above. Specifically, for $\mathcal{R}_0^{vac} < 1$ the disease dies out for any initial condition and for $\mathcal{R}_0^{vac} > 1$ the disease is endemic in the population.

## 3.5 Optimal control

The Theory of Optimal control was developed as an extension of the calculus of variations, by Lev Pontryagin and his collaborators. It is used to determine control strategies that minimize an objective functional, for models where the underlying dynamics are governed by systems of differential equations. It has found wide application in biological models including epidemic models [24–26]. The goal here is to reduce the infected population by means of specific controls, which may appear as time dependent parameters in the model, while minimizing the required resources. The algorithm is implemented by appending an adjoint system of differential equations having terminal conditions along with the original state system. Further, details regarding Optimal Control and adjoint system can be found in [27, 28].

**3.5.1 Optimal vaccine and medication.** We use the theory of optimal control to suggest the 'best' control strategies for the COVID-19 epidemic, which will minimize the total infected numbers while keeping the associated costs low. In the initial phase of the outbreak, only non-pharmaceutical interventions were available to control the epidemic, however by mid 2020, emergency approvals for some promising treatments for the disease were given, followed by emergency approval of vaccines, starting in December 2020.

We consider vaccination and medication measures as possible control strategies for both high and low risk populations. Optimal control theory is used to propose the 'best' control strategy by minimizing a cost functional subject to the differential equation constraints given by the model equations.

Let $\overline{U}$ be the control set defined for the parameters $\tau_L$, $\tau_H$, $\xi_L$ and $\xi_H$ from model (8).

$$\overline{U} = \{\tau_L(t), \tau_H(t), \xi_L(t), \xi_H(t) : 0 \leq \tau_L(t), \tau_H(t), \xi_L(t), \xi_H(t) \leq \overline{\zeta_j}, 0 \leq t \leq T,$$

$$0 < \overline{\zeta_j} \leq 1, j = 1, 2, \cdots, 4\}$$

(13)

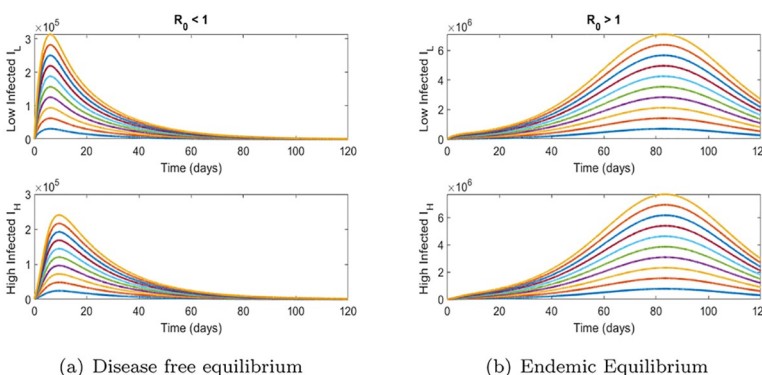

(a) Disease free equilibrium (b) Endemic Equilibrium

**Fig 7. Time series simulations for vaccine model (8).** (a) Disease free equilibrium, (b) Endemic Equilibrium.

Here, $\tau_L(t)$, $\tau_H(t)$, $\xi_L(t)$, $\xi_H(t)$ are Lebesgue measurable and the $\overline{\zeta}_j$, $\forall j = 1, 2, 3, 4$ are positive upper bound of respective control parameters. We wish to minimize the costs incurred due to the burden of disease along with vaccination and medication costs [27].

The functional $\overline{J}$ consists of the infected individuals ($I_L + I_H$) and the nonlinear(quadratic) weighted ($\overline{W}_j$) functions of the control variables $\xi_L$, $\xi_H$, $\tau_L$, $\tau_H$ representing the cost of control.

$$
\begin{aligned}
\overline{J}[\tau_L(t), \tau_H(t), \xi_L(t), \xi_H(t)] &= \int_0^T \left( I_L(t) + I_H(t) + \frac{1}{2} W_1 \tau_L^2(t) + \frac{1}{2} W_2 \tau_H^2(t) \right. \\
&\qquad \left. + \frac{1}{2} W_3 \xi_L^2(t) + \frac{1}{2} W_4 \xi_H^2(t) \right) dt \\
\overline{J}[\tau_L^\star(t), \tau_H^\star(t), \xi_L^\star(t), \xi_H^\star(t)] &= \min_{(\tau_L, \tau_H, \xi_L, \xi_H) \in \overline{U}} \overline{J}[\tau_L(t), \tau_H(t), \xi_L(t), \xi_H(t)]
\end{aligned}
\tag{14}
$$

As described above to calculate the optimal controls an adjoint system is appended to the original model equations (state equations). In our study numerical results are produced using the forward (state system) backward (adjoint system) sweep method with a fourth-order backward Runge-Kutta method.

**Theorem 3.3**. *Given the functional* (14) *subject to the state system* (8), *there exist unique optimal controls* $\tau_L^\star(t)$, $\tau_H^\star(t)$, $\xi_L^\star(t)$, $\xi_H^\star(t)$, (19), *which minimize the functional* $\overline{J}$ *over the control set* $\overline{U}$. *Moreover, there exists feed back control adjoint differential system* (18) *which supports optimizing the vaccination and medication strategies. This adjoint system* (18) *satisfies the transversality conditions* $\{\overline{\Phi}_j(T) = 0, j = 1, 2, \cdots, 12\}$.

*Proof.* Further details are attached in appendix.

**3.5.2 Vaccination and medication strategies.** We now present the optimal vaccination strategy, this minimizes the total infected population over time as well as keeps the cost of control low. We would like to address two issues: (1) Given a maximum possible vaccination rate, how should the vaccination rate vary over time? (2) Should the vaccination strategies differ for the high and low risk groups?

We note that for different proportion of the high risk population the 'best' vaccination strategy is to vaccinate at the highest possible rate initially and then gradually bring down the rate of vaccination. There are two competing effects in our model, the low risk group is assumed to have a higher contact rate and individuals in the high risk group stay infected for a longer period (due to severe infection), both of these tend to increase the total infected population over time. This also makes the vaccination strategy, Fig 8, somewhat insensitive to the high and low risk proportion in the population.

We next consider the optimal medication strategy, Fig 9, the goal is again to study the the time dependent medication rate, and differences if any, in the mediation strategy for high and low risk infected groups.

We note that the medication strategy is insensitive to the proportion of high risk individuals. The optimal strategy is to medicate both high and low risk infected individuals at a high rate throughout the course of the epidemic.

We would like to point out that our goal here was to look at the optimal strategies designed to keep the total infected population at a minimum considering the effects of the high and low risk population proportions. Two other factors may be of importance which we do not consider in this work; the role of mobility and trying to keep the number of fatalities due to disease low, we aim to address these issues in a follow up work. We now sum up our study in the next section.

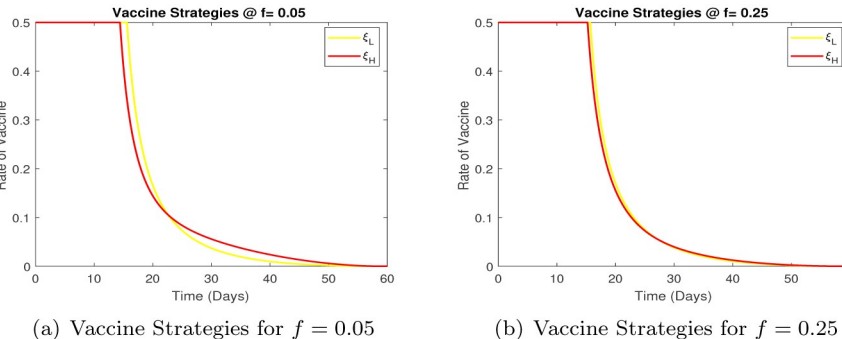

(a) Vaccine Strategies for $f = 0.05$ (b) Vaccine Strategies for $f = 0.25$

**Fig 8. Comparison of vaccine strategies for different values of *f*.** (a) Vaccine Strategies for $f = 0.05$, (b) Vaccine Strategies for $f = 0.25$.

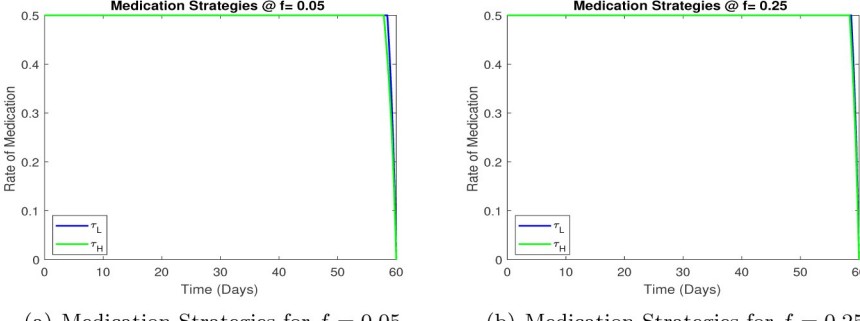

(a) Medication Strategies for $f = 0.05$ (b) Medication Strategies for $f = 0.25$

**Fig 9. Comparison of medication strategies for different values of *f*.** (a) Medication Strategies for $f = 0.05$ (b) Medication Strategies for $f = 0.25$.

## 4 Conclusions

We present and analyze a model for the transmission dynamics of COVID-19. It has been well established that some segments of the population are far more at risk for a more severe infection with a much higher mortality, based on age and presence of co-morbidities. Our model takes this into account by considering two susceptible population subgroups consisting of high and low risk individuals. The transmission within each group is modelled by an extension of the SEIR model, considering first two additional compartments representing quarantined and medicated individuals, as these were the only viable control strategies available during most of 2020 and then vaccination and medication as we now have several vaccines available as well as antiviral therapies. There are two main questions we addressed: (1) does the proportion of the high risk susceptibles in the population lead to a markedly different epidemic curve and (2) if resources are limited, should the available control measures be concentrated on a particular risk group?

- We derive basic properties for the first model using standard dynamical systems techniques. Existence of a disease free state (DFE) and an endemic state (EE) is established. A threshold quantity $\mathcal{R}_0$ is derived such that the DFE is stable whenever $\mathcal{R}_0 < 1$ and unstable otherwise, it is also shown that the EE is stable whenever $\mathcal{R}_0 > 1$.

- Time series plots for the infected population(s) are presented, taking into consideration a varying proportion of susceptibles from the two risk groups. We also plot the cumulative deaths over time for these cases. Our findings show that the difference in numbers of

infected as well as deaths can be explained in part by the difference in proportion of the two risk groups in the susceptible population. Our simulations show that a higher percentage of high risk individuals leads to a higher disease burden and mortality. We also observe that the epidemic peaks earlier for when the proportion of high risk individuals is lower, also contributing to a lower total number of infected.

- We look at contour plots of $\mathcal{R}_0$ to study how it varies with the contact rates of the two classes. To make $\mathcal{R}_0 < 1$ contact rates for both classes need to be brought down, this points towards the rationale of social distancing and mask mandates. We also look at the variation of $\mathcal{R}_0$ with the rate and efficacy of medication, which reinforces the idea that a with more effective medication would require a lower rate of medication for effective disease control.

- We next consider a model with vaccination and medication as control measures. After determining the DFE and EE, we determine $\mathcal{R}_0$, such that the DFE is stable whenever $\mathcal{R}_0 < 1$ and unstable otherwise, it is also shown that the EE is stable whenever $\mathcal{R}_0 > 1$.

- Using ideas from optimal control theory, we then propose optimal vaccination and medication strategies. We need to vaccinate and medicate both groups at the highest possible rate initially and then bring it down over time, there does not seem to be any significant difference in the vaccination strategy based on the proportion of high and low risk individuals. We note here that the goal here was to minimize the total infected population, although this in turn will have the effect of lowering the mortality, we do not consider minimizing the number of deaths directly in this study.

To summarize, we presented a deterministic ODE based compartmental model for the transmission dynamics of COVID-19. We wanted to study the effects of the presence of individuals at high and low risk for severe symptoms and high morbidity in the population. Our findings show that a higher proportion of high risk individuals leads to a higher disease burden and much higher mortality, this has been observed in countries with a high percentage of aging population and/or co-morbidities. Our study also shows that to effectively control the outbreak, available control strategies should be used more or less equally across the two population sub groups, irrespective of their proportion in the total population.

## A Appendix

### A.1 Proof of Lemma 2.1

We can rewrite system (1) as

$$\frac{d\mathbb{X}}{dt} = g(\mathbb{X}) \tag{15}$$

where $\mathbb{X} = (S, Q_L E_L, I_L, M_L, R_L, Q_H E_H, I_H, M_H, R_H)$ and $g(\mathbb{X}) = (g_1(\mathbb{X}), g_2(\mathbb{X}), \cdots, g_{12}(\mathbb{X}))$ represent the RHS of the model (1). It is evident that for all $k = 1, 2, \cdots, 12, g_k(\mathbb{X}) \geq 0$ whenever $\mathbb{X} \in [0, \infty)^{12}$ and $\mathbb{X}_j = 0$. Since total population $N(t) = N_L(t) + N_H(t)$ is positive, $g(\mathbb{X})$ is locally Lipschitz in the set $\mathcal{D}$. It follows from the Theorem A.4 in [29], model (1) shares a positive unique solution in the set $\mathcal{D}$.

Adding all the equations of the model (1)

$$\frac{dN_L}{dt} + \frac{dN_H}{dt} = \pi_L + \pi_H - \mu N_L - \mu N_H - \delta_L I_L - \delta_H I_H - v_L M_L - v_H M_H$$

$$\frac{dN}{dt} = \pi_L + \pi_H - \mu N - \delta_L I_L - \delta_H I_H - v_L M_L - v_H M_H$$

Since $\mathbb{X}_k \geq 0$

$$\frac{dN}{dt} \leq \pi_L + \pi_H - \mu N_H$$

$$\Rightarrow N(t) \leq N(0)e^{-\mu t} + \frac{(\pi_L + \pi_H)}{\mu}\left(1 - e^{-\mu t}\right)$$

Thus if $N(0) \leq \frac{(\pi_L + \pi_H)}{\mu}$, implies $N(t) \leq \frac{(\pi_L + \pi_H)}{\mu}$ for all $t > 0$. Hence the set $\mathcal{D}$ is positively invariant.

## A.2 Proof of Lemma 3.1

The system (8) can be rewritten as

$$\frac{d\mathbb{Y}}{dt} = g(\mathbb{Y}) \tag{16}$$

where $\mathbb{Y} = (S_L, V_L, E_L, I_L, M_L, R_L, S_H, V_H, E_H, I_H, M_H, R_H)$ and $g(\mathbb{Y}) = (g_1(\mathbb{Y}), g_2(\mathbb{Y}), \cdots, g_{12}(\mathbb{Y}))$ are the right hand sides of model (8). It is noticeable that for all $k = 1, 2, \cdots, 12, g_k(\mathbb{Y}) \geq 0$ whenever $\mathbb{Y} \in [0, \infty)^{12}$ and $\mathbb{Y}_j = 0$. As the total population is divided into two sub populations which, $N(t) = N_L(t) + N_H(t)$ is positive, $g(\mathbb{Y})$ is locally Lipschitz in the set $\tilde{\mathcal{D}}$. Using result (Theorem A.4) form [29], model (8) has a positive unique solution in the set $\tilde{\mathcal{D}}$.

Adding all the equations of the model (8)

$$\frac{dN_L}{dt} + \frac{dN_H}{dt} = \pi_L + \pi_H - \mu N_L - \mu N_H - \delta_L I_L - \delta_H I_H - v_L M_L - v_H M_H$$

$$\frac{dN}{dt} = \pi_L + \pi_H - \mu N - \delta_L I_L - \delta_H I_H - v_L M_L - v_H M_H$$

Since $\mathbb{Y}_k \geq 0$

$$\frac{dN}{dt} \leq \pi_L + \pi_H - \mu N_H$$

$$\Rightarrow N(t) \leq N(0)e^{-\mu t} + \frac{(\pi_L + \pi_H)}{\mu}\left(1 - e^{-\mu t}\right)$$

Thus if $N(0) \leq \frac{(\pi_L + \pi_H)}{\mu}$, implies $N(t) \leq \frac{(\pi_L + \pi_H)}{\mu}$ for all $t > 0$. Hence the set $\tilde{\mathcal{D}}$ is positively invariant.

## A.3 Proof of Lemma 3.3

*Proof.* The Hamiltonian can be written as

$$\overline{H} = I_L + I_H + \frac{1}{2}W_1\tau_L^2 + \frac{1}{2}W_2\tau_H^2 + \frac{1}{2}W_3\xi_L^2 + \frac{1}{2}W_4\xi_H^2 + \sum_{j=1}^{12}\overline{\Phi}_j f_j$$

where the $f_j$, $j = 1, 2, \cdots, 12$ are right hand sides of the model (8). It can be easily shown that the Integrand $\overline{J}(\cdot)$ is convex with respect to the control variables defined as $\tau_L, \tau_H, \xi_L, \xi_H$. Lemma (3.1) guarantees that state system solutions are positive and bounded above by $N(t) \leq \frac{\pi_L + \pi_H}{\mu}, \forall t > 0$. Also, The model (8) follows Lipschitz property with respect to the state

variables. Combining the above three properties i.e., Convexity of the Integrand $\overline{J}$, bounded-ness of state system solutions with Lipsctiz property ensures us the existence of the optimal solution of the control variables over the set $\overline{U}$ [30]. Using Pontryain's Maximum principle conditions, the adjoint system can be written as

$$
\begin{aligned}
\frac{d\overline{\Phi_1}}{dt} &= -\frac{\partial \overline{H^*}}{\partial S_L}, \overline{\Phi_1}(T) = 0 \\
\frac{d\overline{\Phi_2}}{dt} &= -\frac{\partial \overline{H^*}}{\partial Q_L}, \overline{\Phi_2}(T) = 0 \\
\vdots \qquad \vdots \qquad \vdots \\
\frac{d\overline{\Phi_{12}}}{dt} &= -\frac{\partial \overline{H^*}}{\partial R_H}, \overline{\Phi_{12}}(T) = 0
\end{aligned}
\tag{17}
$$

$$
\begin{aligned}
\frac{d\overline{\Phi_1}}{dt} &= \overline{\Phi_1}(\mu + \lambda_L + \xi_L) - \overline{\Phi_3}\lambda_L - \overline{\Phi_2}\xi_L \\
\frac{d\overline{\Phi_2}}{dt} &= -\overline{\Phi_1}\omega_L + \overline{\Phi_2}(\mu + \omega_L - (\epsilon - 1)\lambda_L) + (\epsilon - 1)\overline{\Phi_3}\lambda_L \\
\frac{d\overline{\Phi_3}}{dt} &= \overline{\Phi_3}(\mu + \sigma_L) - \overline{\Phi_4}\sigma_L \\
\frac{d\overline{\Phi_4}}{dt} &= \frac{\overline{\Phi_7}S_H\beta_{LH}}{N_H} - \overline{\Phi_9}\left(\frac{S_H\beta_{LH}}{N_H} + \frac{(1 - \epsilon)V_H\beta_{LH}}{N_H}\right) + \frac{(1 - \epsilon)\overline{\Phi_8}V_H\beta_{LH}}{N_H} \\
&\quad + \overline{\Phi_4}(\mu\gamma_L + \delta_L + \tau_L) - \overline{\Phi_6}\gamma_L + \frac{\overline{\Phi_1}\beta_L S_L}{N_L} - \overline{\Phi_3}\left(\frac{\beta_L S_L}{N_L} + \frac{(1 - \epsilon)\beta_L V_L}{N_L}\right) \\
&\quad + \frac{(1 - \epsilon)\overline{\Phi_2}\beta_L V_L}{N_L} - \overline{\Phi_5}\tau_L - 1 \\
\frac{d\overline{\Phi_5}}{dt} &= \overline{\Phi_5}(\mu + \kappa_L + v_L) - \overline{\Phi_6}\kappa_L + \frac{\overline{\Phi_1}\beta_L S_L \phi_L}{N_L} \\
&\quad + \frac{(1 - \epsilon)\overline{\Phi_2}\beta_L V_L \phi_L}{N_L} - \overline{\Phi_3}\left(\frac{\beta_L S_L \phi_L}{N_L} + \frac{(1 - \epsilon)\beta_L V_L \phi_L}{N_L}\right) \\
\frac{d\overline{\Phi_6}}{dt} &= \mu\overline{\Phi_6} \\
\frac{d\overline{\Phi_7}}{dt} &= \overline{\Phi_7}(\lambda_H + \xi_H + \mu) - \overline{\Phi_9}\lambda_H - \overline{\Phi_8}\xi_H \\
\frac{d\overline{\Phi_8}}{dt} &= -\overline{\Phi_7}\omega_H + \overline{\Phi_8}(\omega_H - (\epsilon - 1)\lambda_H + \mu) + (\epsilon - 1)\overline{\Phi_9}\lambda_H \\
\frac{d\overline{\Phi_9}}{dt} &= \overline{\Phi_9}(\sigma_H + \mu) - \overline{\Phi_{10}}\sigma_H \\
\frac{d\overline{\Phi_{10}}}{dt} &= -1 + \overline{\Phi_{10}}(\gamma_H + \delta_H + \tau_H + \mu) - \overline{\Phi_{12}}\gamma_H + \frac{\overline{\Phi_7}\beta_H S_H}{N_H} \\
&\quad - \overline{\Phi_9}\left(\frac{\beta_H S_H}{N_H} + \frac{(1 - \epsilon)\beta_H V_H}{N_H}\right) - \overline{\Phi_{11}}\tau_H + \frac{(1 - \epsilon)\overline{\Phi_8}\beta_H V_H}{N_H} + \frac{\overline{\Phi_1}\beta_{HL}S_L}{N_L} \\
&\quad - \overline{\Phi_3}\left(\frac{\beta_{HL}S_L}{N_L} + \frac{(1 - \epsilon)\beta_{HL}V_L}{N_L}\right) + \frac{(1 - \epsilon)\overline{\Phi_2}\beta_{HL}V_L}{N_L} \\
\frac{d\overline{\Phi_{11}}}{dt} &= \overline{\Phi_{11}}(\kappa_H + v_H + \mu) - \overline{\Phi_{12}}\kappa_H + \frac{\overline{\Phi_7}\beta_H S_H \phi_H}{N_H} \\
&\quad - \overline{\Phi_9}\left(\frac{\beta_H S_H \phi_H}{N_H} + \frac{(1 - \epsilon)\beta_H V_H \phi_H}{N_H}\right) + \frac{(1 - \epsilon)\overline{\Phi_8}\beta_H V_H \phi_H}{N_H} \\
\frac{d\overline{\Phi_{12}}}{dt} &= \mu\overline{\Phi_{12}}
\end{aligned}
\tag{18}
$$

The optimal conditions can be written as

$$\frac{\partial \overline{H^*}}{\partial \tau_L} = 0 \Rightarrow \tau_L = \left( \frac{(\overline{\Phi_4} - \overline{\Phi_5})I_L}{W_1} \right)$$

$$\frac{\partial \overline{H^*}}{\partial \tau_H} = 0 \Rightarrow \tau_H = \left( \frac{(\overline{\Phi_{10}} - \overline{\Phi_{11}})I_H}{W_2} \right)$$

$$\frac{\partial \overline{H^*}}{\partial \xi_L} = 0 \Rightarrow \xi_L = \left( \frac{(\overline{\Phi_1} - \overline{\Phi_2})S_L}{W_3} \right)$$

$$\frac{\partial \overline{H^*}}{\partial \xi_H} = 0 \Rightarrow \xi_H = \left( \frac{(\overline{\Phi_7} - \overline{\Phi_8})S_H}{W_4} \right)$$

Since the control variables are bounded in set in $\overline{U}$, The control variables are updated according the max limits in set $\overline{U}$ by $\overline{\zeta_i}$'s. The optimal controls becomes:

$$
\begin{aligned}
\tau_L^\star(t) &= \min \left[ \overline{\zeta_1}, \, \max \left( 0, \frac{(\overline{\Phi_4} - \overline{\Phi_5})I_L}{W_1} \right) \right] \\
\tau_H^\star(t) &= \min \left[ \overline{\zeta_2}, \, \max \left( 0, \frac{(\overline{\Phi_{10}} - \overline{\Phi_{11}})I_H}{W_2} \right) \right] \\
\xi_L^\star(t) &= \min \left[ \overline{\zeta_3}, \, \max \left( 0, \frac{(\overline{\Phi_1} - \overline{\Phi_2})S_L}{W_3} \right) \right] \\
\xi_H^\star(t) &= \min \left[ \overline{\zeta_4}, \, \max \left( 0, \frac{(\overline{\Phi_7} - \overline{\Phi_8})S_H}{W_4} \right) \right]
\end{aligned}
\tag{19}
$$

The uniqueness of optimal controls is followed from the uniqueness of the optimal uniqueness of the optimality systems (state and adjoint).

## Author Contributions

**Conceptualization:** Adnan Khan, Mohsin Ali, Mudassar Imran.

**Formal analysis:** Mohsin Ali, Wizda Iqbal, Mudassar Imran.

**Software:** Mohsin Ali, Wizda Iqbal.

**Supervision:** Adnan Khan, Mudassar Imran.

**Writing – original draft:** Mohsin Ali, Wizda Iqbal.

**Writing – review & editing:** Adnan Khan, Mudassar Imran.

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
