## [Decision Letter · Decision Letter 0]

27 Jul 2021

PONE-D-21-22554

Effect of High and Low Risk Susceptible in the Transmission Dynamics of COVID-19 and Control Strategies

PLOS ONE

Dear Dr. Imran,

Thank you for submitting your manuscript to PLOS ONE. After careful consideration, we feel that it has merit but does not fully meet PLOS ONE’s publication criteria as it currently stands. Therefore, we invite you to submit a revised version of the manuscript that addresses the points raised during the review process.

We look forward to receiving your revised manuscript.

Kind regards,

Maria Alessandra Ragusa, PhD Professor

Academic Editor

PLOS ONE

Journal Requirements:

4. Please ensure that you refer to Figure 2,3,4,5,8 and 9 in your text as, if accepted, production will need this reference to link the reader to the figure.

5. We note you have included a table to which you do not refer in the text of your manuscript. Please ensure that you refer to Table 1 in your text; if accepted, production will need this reference to link the reader to the Table.

Additional Editor Comments:

Dear corresponding author,

the paper must be revised, then upload it again.

Best regards.

Reviewers' comments:

Reviewer's Responses to Questions

**Comments to the Author**

1. Is the manuscript technically sound, and do the data support the conclusions?

Reviewer #1: Yes

Reviewer #2: Partly

2. Has the statistical analysis been performed appropriately and rigorously? 

Reviewer #1: Yes

Reviewer #2: Yes

3. Have the authors made all data underlying the findings in their manuscript fully available?

Reviewer #1: Yes

Reviewer #2: No

4. Is the manuscript presented in an intelligible fashion and written in standard English?

Reviewer #1: Yes

Reviewer #2: No

5. Review Comments to the Author

Reviewer #1: The work is characterized by modernity, and the model presented in the paper is new and serves a vital topic that our world suffers from these days, which is the emerging Corona virus (COVID-19).

I would be grateful to recommend this paper for publication in PLOS ONE.

Reviewer #2: 1-In particular, the main contributions should be pointed out.

2- The author does not adequately substantiates the importance of studying models

. What is its relevance applied? what phenomena modeled?

3- proof of lemma 2.2 must be added

4- Check the manuscript carefully for typos and grammatical errors.

5- If possible compare the results with some known methods and list in tabular form

6- Figures must be added more dettails

5- update the references by adding the following references:

-TRANSMISSION DYNAMICS AND CONTROL STRATEGIES OF COVID-19 IN WUHAN, CHINA, https://doi.org/10.1142/S0218339020500096

- Fractional Stochastic

Models for COVID-19: Case Study of Egypt, Results in Physics (2021): 104018.

- Models for COVID-19 Daily Confirmed Cases in Different Countries, Mathematics 9, 2021,

659. https://doi.org/10.3390/math9060659.

6. PLOS authors have the option to publish the peer review history of their article (what does this mean?). If published, this will include your full peer review and any attached files.

Reviewer #1: **Yes: **Mohamed I. Abbas

Reviewer #2: No

---

## [Author Response · Author response to Decision Letter 0]

24 Aug 2021

Response Letter for the PLOS revision

We would like to thank the reviewer for the speedy turnaround and valuable comments. Their feedback has allowed us to improve the manuscript. 

Editors comments: 

1. Please ensure that your manuscript meets PLOS ONE's style requirements, including those for file naming

 The manuscript is converted to the PLOS ONE format, which meets the style requirement.

2. In your Data Availability statement, you have not specified where the minimal data set underlying the results described in your manuscript can be found.

The data that support the findings of this study are openly available in at 

a. COVID-19 Data Repository by the Center for Systems Science and Engineering (CSSE) at Johns Hopkins University [https://github.com/CSSEGISandData/COVID-19]

b. https://unstats.un.org/unsd/demographic-social/products/dyb/index.cshtml

3. Please note that in order to use the direct billing option the corresponding author must be affiliated with the chosen institute.

Noted with thanks. 

4. Please ensure that you refer to Figure 2,3,4,5,8 and 9 in your text as, if accepted, production will need this reference to link the reader to the figure.

The figures are referred to in the text, with the style format. 

5. We note you have included a table to which you do not refer in the text of your manuscript. Please ensure that you refer to Table 1 in your text

This has been updated. 

Reviewer comments

1. In particular, the main contributions should be pointed out.

The conclusions section has been updated, making the contributions clearer. 

2. The author does not adequately substantiates the importance of studying models

. What is its relevance applied? what phenomena modeled?

This has been incorporated in the Introduction section. 

3. proof of lemma 2.2 must be added

Proof of lemma 2.2 has been added

4. Check the manuscript carefully for typos and grammatical errors.

The manuscript has been updated in light of the comment. 

5. 5- If possible compare the results with some known methods and list in tabular form

Our model explores a hypothesis which to our knowledge, has not be quantitatively done for COVID-19.

6- Figures must be added more details

Figures are referred to in the text with details.

6. update the references by adding the following references:

-TRANSMISSION DYNAMICS AND CONTROL STRATEGIES OF COVID-19 IN WUHAN, CHINA, https://doi.org/10.1142/S0218339020500096

- Fractional Stochastic

Models for COVID-19: Case Study of Egypt, Results in Physics (2021): 104018.

- Models for COVID-19 Daily Confirmed Cases in Different Countries, Mathematics 9, 2021,

659. https://doi.org/10.3390/math9060659.

The references have been added in the Introduction section.

---

## [Editor Report · Decision Letter 1]

31 Aug 2021

Effect of High and Low Risk Susceptibles in the Transmission Dynamics of COVID-19 and Control Strategies

PONE-D-21-22554R1

Dear Dr. Imran,

We’re pleased to inform you that your manuscript has been judged scientifically suitable for publication and will be formally accepted for publication once it meets all outstanding technical requirements.

Kind regards,

Maria Alessandra Ragusa, PhD Professor

Academic Editor

PLOS ONE

Additional Editor Comments (optional):

The revised version is now ready fpr publication. Best regards.
---

## [Editor Report · Acceptance letter]

6 Sep 2021

PONE-D-21-22554R1 

Effect of High and Low Risk Susceptibles in the Transmission Dynamics of COVID-19 and Control Strategies 

Dear Dr. Imran:

I'm pleased to inform you that your manuscript has been deemed suitable for publication in PLOS ONE. Congratulations! Your manuscript is now with our production department. 

Kind regards, 

on behalf of

Dr. Maria Alessandra Ragusa 

Academic Editor

PLOS ONE